# UNSUPERVISED VISUALIZATION OF IMAGE DATASETS USING CONTRASTIVE LEARNING

**Jan Niklas Böhm, Philipp Berens & Dmitry Kobak**
University of Tübingen, Germany
{jan-niklas.boehm,philipp.berens,dmitry.kobak}@uni-tuebingen.de

## ABSTRACT

Visualization methods based on the nearest neighbor graph, such as $t$-SNE or UMAP, are widely used for visualizing high-dimensional data. Yet, these approaches only produce meaningful results if the nearest neighbors themselves are meaningful. For images represented in pixel space this is not the case, as distances in pixel space are often not capturing our sense of similarity and therefore neighbors are not semantically close. This problem can be circumvented by self-supervised approaches based on contrastive learning, such as SimCLR, relying on data augmentation to generate implicit neighbors, but these methods do not produce two-dimensional embeddings suitable for visualization. Here, we present a new method, called $t$-SimCNE, for unsupervised visualization of image data. $t$-SimCNE combines ideas from contrastive learning and neighbor embeddings, and trains a parametric mapping from the high-dimensional pixel space into two dimensions. We show that the resulting 2D embeddings achieve classification accuracy comparable to the state-of-the-art high-dimensional SimCLR representations, thus faithfully capturing semantic relationships. Using $t$-SimCNE, we obtain informative visualizations of the CIFAR-10 and CIFAR-100 datasets, showing rich cluster structure and highlighting artifacts and outliers.

## 1 INTRODUCTION

As many research fields are producing ever larger and more complex datasets, data visualization methods have become important in many scientific and practical applications (Becht et al., 2019; Diaz-Papkovich et al., 2019; Kobak & Berens, 2019; Schmidt, 2018). Such methods allow a concise summary of the entire dataset, displaying a high-dimensional dataset as a 2D *embedding*. This low-dimensional representation is often convenient for data exploration, highlighting clusters and relationships between them. In practice, most useful are *neighbor embedding* methods, such as $t$-SNE (van der Maaten & Hinton, 2008) and UMAP (McInnes et al., 2018), that aim to preserve nearest neighbors from the high-dimensional space when optimizing the layout in 2D.

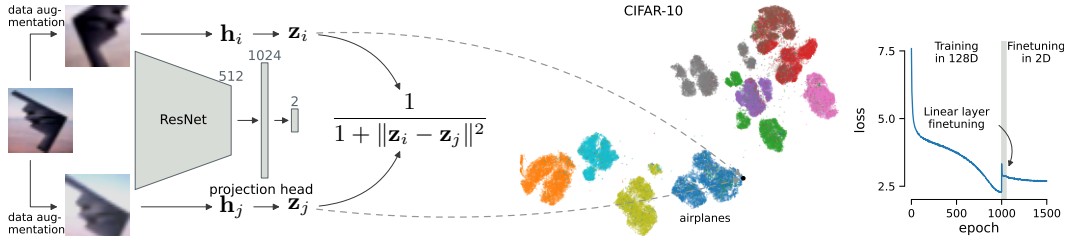

Figure 1: *Left: t-SimCNE.* Two augmentations of the same image are fed through the same ResNet and fully-connected projection head to get representations $\mathbf{z}_i$ and $\mathbf{z}_j$. The loss function pushes $\mathbf{z}_i$ and $\mathbf{z}_j$ together to maximize their Cauchy similarity. *Middle: Embedding of CIFAR-10.* The dashed arrows point to the locations of $\mathbf{z}_i$ and $\mathbf{z}_j$ from the left. *Right: Training loss.* The optimization consists of three stages: (1) pre-training with a 128D output for 1000 epochs; (2) fine-tuning only the 2D readout layer for 50 epochs; and (3) fine-tuning the entire network for 450 epochs.

Unfortunately, for image datasets, nearest neighbors computed using the Euclidean metric in pixel space are typically not worth preserving. Although $t$-SNE works well on very simple image datasets such as MNIST (van der Maaten & Hinton, 2008, Figure 2a), the approach fails when considering more natural image datasets such as CIFAR-10/100 (Supp. Fig. A.1). To create 2D embeddings for images, new visualization approaches are required, which use different notions of similarity.

Here, we provide such a method based on the contrastive learning framework. Contrastive learning is currently the state-of-the-art approach to unsupervised learning in computer vision (Hadsell et al., 2006). The contrastive learning method SimCLR (Chen et al., 2020) uses image transformations to create two views of each image and then optimizes a convolutional neural network so that the two views always stay close together in the resulting representation. While this method performs very well in benchmarks — such as linear or $k$NN classification accuracy, — the computed representation is typically high-dimensional (e.g. 128-dimensional), hence not suitable for visualization.

We extend the SimCLR framework to directly optimize a 2D embedding. Taking inspiration from $t$-SNE, we use the Euclidean distance and the Cauchy ($t$-distribution) kernel to measure similarity in 2D. While using 2D instead of 128D output may not seem like a big step, we show that optimizing the resulting architecture is challenging. We develop an efficient training strategy to overcome these challenges, and only then are able to achieve satisfactory visualizations. We call the resulting method $t$-SimCNE (Fig. 1) and show that it yields meaningful and useful embeddings of CIFAR-10 and CIFAR-100 datasets (Krizhevsky, 2009).

Our code is available at `github.com/berenslab/t-simcne` (see `iclr2023` branch).

## 2 RELATED WORK

Neighbor embeddings (NE) have a rich history dating back to locally linear embedding (Roweis & Saul, 2000) and stochastic neighbor embedding (SNE; Hinton & Roweis, 2003). They became widely used after the introduction of the Cauchy kernel into the SNE framework (van der Maaten & Hinton, 2008) and after efficient approximations became available (van der Maaten, 2014; Linderman et al., 2019). A number of algorithms based on that framework, such as LargeVis (Tang et al., 2016), UMAP (McInnes et al., 2018), and TriMap (Amid & Warmuth, 2019) have been developed and got widespread adoption in recent years in a variety of application fields. All of them are closely related to SNE (Böhm et al., 2022; Damrich et al., 2023) and rely on the $k$NN graph of the data.

NE algorithms have been used to visualize latent representations of neural networks trained in a supervised setting (e.g. Karpathy, 2014; Mnih et al., 2015). This approach is, however, unsuitable for data exploration as it is supervised. NE algorithms can also be applied to an unsupervised (also known as self-supervised) representation of a dataset obtained with SimCLR, or to a representation obtained with a neural network pre-trained on a generic image classification task such as ImageNet (e.g. Narayan et al., 2015). The downside is that these approaches would not yield a parametric mapping to 2D. In this work, we are interested in an unsupervised but parametric mapping that allows embedding out-of-sample points. See Discussion for further considerations.

Conceptual similarity between SimCLR and $t$-SNE has recently been pointed out by Damrich et al. (2023). The authors suggest interpreting $k$NN graph edges as data augmentations, and show that $t$-SNE can also be optimized using the InfoNCE loss (van den Oord et al., 2018) used by SimCLR, and/or using a parametric mapping. Equivalently, one can think of SimCLR as a parametric SNE that samples edges from an unobservable neighbor graph. We were motivated by this connection when developing $t$-SimCNE. Further motivation comes from a recently described phenomenon called *dimensional collapse* (Jing et al., 2022; Tian, 2022), which suggests that there is redundant information in the output of SimCLR. Hence we reasoned that it should be possible to achieve a good representation even with drastically reduced output dimensionality.

Two closely related works appeared during preparation of this manuscript: Zang et al. (2022) suggest an architecture similar to SimCLR for 2D embeddings, but use a more complicated setup to impose 'local flatness' and, judging from their figures, obtain qualitatively worse embeddings of CIFAR datasets than we do (we were unable to quantitatively benchmark their method). Hu et al. (2023) suggest to use the Cauchy kernel in the SimCLR framework (calling it $t$-SimCLR), but in terms of 2D visualization, obtain worse results than we do (Hu et al., 2023, Fig. B.11 shows CIFAR-10, reported $k$NN accuracy 57% vs. our 89%).

## 3 CONTRASTIVE LEARNING FOR VISUALIZATION

### 3.1 SIMCLR OVERVIEW

SimCLR relies on creating two views of each data sample using random data augmentations (Fig. 1). For image data, this means that the image is randomly cropped, flipped, and so on. The method feeds a mini-batch containing pairs of augmented images through a ResNet to obtain a representation of the images. The parameters of the network are optimized such that the paired images are placed close to each other in the output space, while keeping all other images further apart.

In mathematical terms, the loss function used by SimCLR, known as InfoNCE loss (van den Oord et al., 2018), is defined as

$$\ell_{\text{SimCLR}}(i, j) = -\log \frac{\exp\big(\operatorname{sim}(\mathbf{z}_i, \mathbf{z}_j)/\tau\big)}{\sum_{k \neq i}^{2b} \exp\big(\operatorname{sim}(\mathbf{z}_i, \mathbf{z}_k)/\tau\big)} \tag{1}$$

$$= -\operatorname{sim}(\mathbf{z}_i, \mathbf{z}_j)/\tau + \log \sum_{k \neq i}^{2b} \exp\big(\operatorname{sim}(\mathbf{z}_i, \mathbf{z}_k)/\tau\big). \tag{2}$$

Here, indices $i$ and $j$ correspond to two data augmentations of the same original image, and $\mathbf{z}$ denotes network output. For batch size $b$, there are $2b$ samples in the batch because each sample is augmented twice. The similarity function function used by SimCLR is the cosine similarity: $\operatorname{sim}(\mathbf{x}, \mathbf{y}) = \mathbf{x}^\top \cdot \mathbf{y}/(\|\mathbf{x}\| \cdot \|\mathbf{y}\|)$. We follow Chen et al. (2020) in using $\tau = 0.5$.

The SimCLR cost function is intimately related to the SNE loss. Note that the cosine distance $\operatorname{cosdist}(\mathbf{x}, \mathbf{y}) = 1 - \operatorname{sim}(\mathbf{x}, \mathbf{y})$ is equal to the half of the squared Euclidean distance between unit-normalized $\mathbf{x}$ and $\mathbf{y}$. Let $\tilde{\mathbf{a}} = \mathbf{a}/\|\mathbf{a}\|$. Then

$$\exp\big(\operatorname{sim}(\mathbf{z}_i, \mathbf{z}_j)/\tau\big) = \exp\left(\frac{1 - \operatorname{cosdist}(\mathbf{z}_i, \mathbf{z}_j)}{\tau}\right) = \operatorname{const} \cdot \exp\left(-\frac{\|\tilde{\mathbf{z}}_i - \tilde{\mathbf{z}}_j\|^2}{2\tau}\right), \tag{3}$$

which is precisely the expression used by SNE (a Gaussian kernel of the Euclidean distance) to measure similarity between embedding vectors (Hinton & Roweis, 2003). As shown by Damrich et al. (2023), SNE can also be optimized using the InfoNCE loss and mini-batch training; in the SNE setup, $i$ and $j$ are two points connected by a $k$NN graph edge.

The loss functions of both SimCLR and $t$-SNE give rise to an attractive force between similar points $i$ and $j$, enforced by the InfoNCE numerator in Eq. (1), and a repulsion between all points, enforced by the denominator. Wang & Isola (2020) called this the *alignment* and the *uniformity* aspects of the loss. In the neighbor embedding literature, this is referred to as attractive and repulsive forces acting on the embedding points (Böhm et al., 2022; Damrich & Hamprecht, 2021; Wang et al., 2021).

### 3.2 $t$-SIMCNE LOSS FUNCTION FOR 2D VISUALIZATION

To achieve our goal of adapting SimCLR to create 2D embeddings of images, we reduce the dimensionality of the linear output layer from 128 to 2 (Fig. 1). This change, however, requires changing the similarity function as well, as using cosine similarity would effectively constrain the embedding to the unit circle $S^1$, which is not suitable for data visualization.

Instead, we remove the normalization from the last expression in Eq. (3), allowing the output $\mathbf{z}$ vectors to lie anywhere in $\mathbb{R}^2$. In other words, we replace the cosine distance with the Euclidean distance, which is a natural choice for measuring distance between points in a 2D embedding. While one could still use Gaussian kernel to transform Euclidean distance into a similarity as in Eq. (3), we follow $t$-SNE's example (van der Maaten & Hinton, 2008) and use the Cauchy ($t$-distribution with one degree of freedom) kernel instead, as the heavy-tailed Cauchy kernel reduces the 'crowding problem' that SNE had with the Gaussian kernel (van der Maaten & Hinton, 2008). We call the resulting method $t$-SimCNE.

Table 1: Model performance on the CIFAR-10 dataset. The columns are: model type (see text), dimensionality of $Z$, the total number of training epochs, linear classification accuracy in $H$, $k$NN classification accuracy in $Z$ ($k = 15$; see Fig. A.3 for $k \in [1, 30]$), the final loss value, training time. Standard deviations correspond to three runs with different random seeds (for the loss, the standard deviation was always smaller than 0.05). Each experiment was run on a single GeForce RTX 2080 Ti GPU (the 5000 epoch experiment was run on a V100 GPU).

| # | Model | dim | Epochs | Linear in $H$ | $k$NN in $Z$ | Loss | Time (hr.) |
|---|-------|-----|--------|---------------|--------------|------|------------|
| 1 | Cosine (SimCLR) | 128 | 1000 | $93.1 \pm 0.1\%$ | $91.1 \pm 0.2\%$ | 5.8 | $12.0 \pm 0.1$ |
| 2 | Cosine | 3 | 1000 | $87.6 \pm 0.3\%$ | $74.0 \pm 1.4\%$ | 6.3 | $12.3 \pm 0.4$ |
| 3 | Euclidean | 128 | 1000 | $90.7 \pm 0.3\%$ | $90.1 \pm 0.2\%$ | 2.3 | $12.9 \pm 1.6$ |
| 4 | Euclidean | 2 | 1000 | $84.6 \pm 0.3\%$ | $80.0 \pm 0.2\%$ | 3.7 | $11.9 \pm 0.1$ |
| 5 | Euclidean | 2 | 5000 | $88.7 \pm 0.2\%$ | $87.0 \pm 0.5\%$ | 2.9 | $65.4 \pm 0.2$ |
| 6 | Cos.→Euclidean | 2 | 1500 | $93.0 \pm 0.3\%$ | $90.1 \pm 0.4\%$ | 3.2 | $17.5 \pm 0.1$ |
| 7 | **Eucl.→Euclidean** | **2** | **1500** | **$90.6 \pm 0.2\%$** | **$89.4 \pm 0.4\%$** | **2.7** | **$18.9 \pm 2.3$** |
| 8 | Eucl.→Euclidean | 2 | 1000 | $90.3 \pm 0.3\%$ | $88.3 \pm 0.4\%$ | 2.9 | $11.8 \pm 0$ |
| 9 | Eucl.→Euclidean | 2 | 500 | $88.7 \pm 0.1\%$ | $85.8 \pm 0.1\%$ | 3.3 | $6.0 \pm 0$ |

We denote the Euclidean distance between $\mathbf{z}_i$ and $\mathbf{z}_j$ as $d_{ij} = \|\mathbf{z}_i - \mathbf{z}_j\|$. The corresponding Cauchy similarity is $1/(1 + d_{ij}^2)$. With that, we define the $t$-SimCNE loss as

$$\ell_{t\text{-SimCNE}}(i, j) = -\log \frac{1/(1 + d_{ij}^2)}{\sum_{k \neq i}^{2b} 1/(1 + d_{ik}^2)} \tag{4}$$

$$= -\log \frac{1}{1 + d_{ij}^2} + \log \sum_{k \neq i}^{2b} \frac{1}{1 + d_{ik}^2} . \tag{5}$$

We will refer to the SimCLR loss as 'cosine' loss and to the $t$-SimCNE loss as 'Euclidean' loss. Note that the numerical loss values between them are not directly comparable, because the minimum of the Euclidean loss is zero, whereas the cosine loss, Eq. (2), is bounded from below by

$$\ell_{\text{SimCLR}}^*(i, j) = -1/\tau + \log \left( \exp(1/\tau) + \sum_{k \neq i}^{2b-1} \exp(-1/\tau) \right)$$

$$= -1/\tau + \log \left( \exp(1/\tau) + (2b - 2) \cdot \exp(-1/\tau) \right) . \tag{6}$$

which is $\sim 3.65$ for $b = 1024$ and $\tau = 0.5$.

### 3.3 IMPLEMENTATION, DATASETS, AND PERFORMANCE METRICS

We used our own PyTorch (Paszke et al., 2019, version 1.12.1) implementation of SimCLR. As the backbone, we used a ResNet18 (He et al., 2016), which has 512-dimensional output. We reduced the kernel size in the first convolutional layer of the ResNet18 from $7 \times 7$ to $3 \times 3$ as in Chen et al. (2020). For the fully-connected projection head we used one hidden ReLU layer with 1024 units, and linear output layer with 128 units. We optimized the network for 1000 epochs using SGD with momentum 0.9. The initial learning rate was $0.03 \cdot b/256 = 0.12$, with linear warm-up over ten epochs (from 0 to 0.12) and cosine annealing (Loshchilov & Hutter, 2017) down to 0 for the remaining epochs. We used batch size $b = 1024$ and the same set of data augmentations as in Chen et al. (2020).

We used CIFAR-10 and CIFAR-100 datasets (Krizhevsky, 2009) for all our experiments. Each dataset consists of $n = 60\,000$ colored and labeled $32 \times 32$ images. CIFAR-10 has 10 classes (Fig. A.2), while CIFAR-100 has 100 classes grouped into 20 superclasses. We used the dataset classes `CIFAR10` and `CIFAR100` provided by the `torchvision` package.

To quantitatively assess the embedding quality, we not only report loss values but also the test-set $k$NN accuracy in the projection head output space $Z$ as our main performance metric. Note that in the contrastive learning literature (e.g. Chen et al., 2020), representation quality is usually assessed

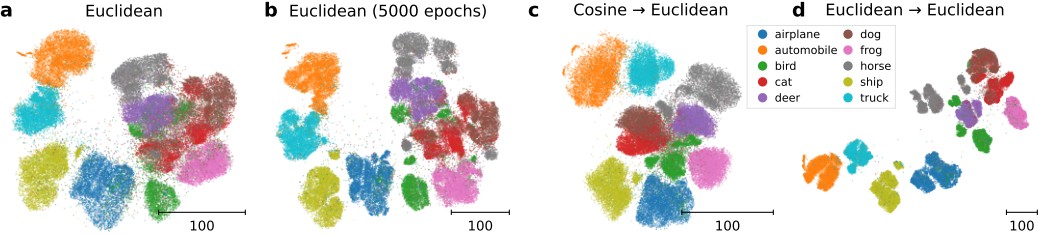

Figure 2: Different training strategies for $t$-SimCNE on CIFAR-10. *(a)* Optimizing the 2D Euclidean loss for 1000 epochs. *(b)* Optimizing the 2D Euclidean loss for 5000 epochs. *(c)* Pretraining with cosine loss in 128D and fine-tuning with Euclidean loss in 2D. *(d)* Pretraining with Euclidean loss in 128D and fine-tuning with Euclidean loss in 2D.

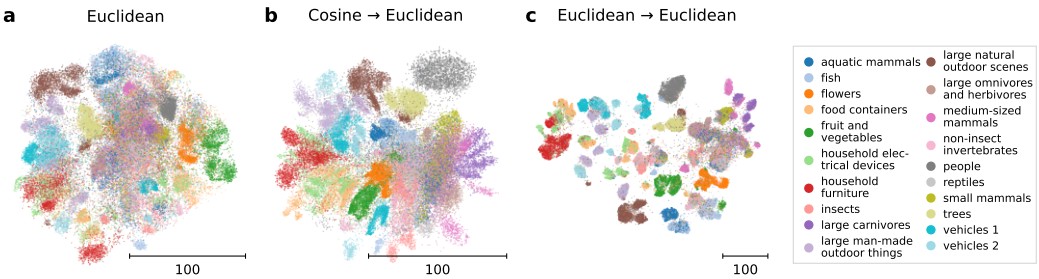

Figure 3: Different training strategies for $t$-SimCNE on CIFAR-100. The colors correspond to 20 superclasses and not to the fine-grained labels. *(a)* Optimizing the 2D Euclidean loss for 1000 epochs. *(b)* Pretraining with cosine loss in 128D and fine-tuning with Euclidean loss in 2D. *(c)* Pretraining with Euclidean loss in 128D and fine-tuning with Euclidean loss in 2D.

via linear classification accuracy in the ResNet output space $H$. In the $Z$ space, we prefer to use the $k$NN classifier, as the 2D embedding can be considered good even if classes are separable but not linearly separable. For $k$NN accuracy, we used the scikit-learn (Pedregosa et al., 2011) implementation with $k = 15$ (any $k \in [1, 30]$ gave qualitatively the same results, Fig. A.3). For the linear accuracy, we used the `LogisticRegression` class from scikit-learn, with the SAGA solver (Defazio et al., 2014) and no penalty. For CIFAR-10 (CIFAR-100), our SimCLR implementation achieved 93% (67%) test-set linear accuracy in $H$ (Tables 1 and A.1, line 1) which is state of the art for SimCLR with ResNet18 (e.g. Chen & He, 2021, Appendix D and da Costa et al., 2022, Table 1).

We heavily used Matplotlib 3.6.0 (Hunter, 2007), NumPy 1.23.1 (Harris et al., 2020), and openTSNE 0.6.2 (Poličar et al., 2019), which, in turn, uses Annoy (Bernhardsson, 2013).

### 3.4 $t$-SIMCNE REQUIRES DIMENSIONALITY ANNEALING FOR OPTIMAL RESULTS

We used the setup described in the previous section to train $t$-SimCNE on the entire CIFAR-10 dataset. However, we found that naïve training of 2D $t$-SimCNE resulted in a suboptimal embedding layout (Fig. 2a) and achieved only 80% $k$NN accuracy (Table 1, line 4).

When we increased the number of training epochs from 1000 to 5000, the embedding improved (Fig. 2b), with $k$NN accuracy reaching 87% (Table 1, line 5). However, 5000 epochs took a long time (almost three GPU days) so this approach is not very practical. Moreover, we felt the embedding was still suboptimal, with many clusters seemingly unable to break apart from each other.

We noticed that in both cases above, the linear accuracy in $H$ was markedly lower than with standard SimCLR (85% and 89% vs. 93%, cf. Table 1, lines 4, 5, and 1). We reasoned that it may be beneficial to pretrain the model with 128D output, achieving high-quality representation in $H$, and then fine-tune the model with the 2D output. This can be seen as 'dimensionality annealing'. Moving from 128D output to 2D requires changing the linear output layer in the projection head. For this, we

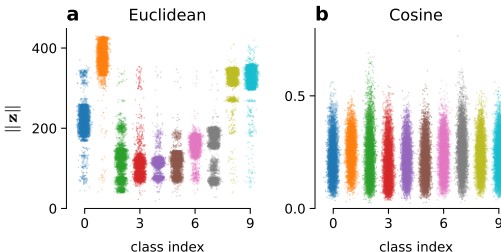

Figure 4: $L_2$ norms of the representation in 128-dimensional $Z$ space after training on CIFAR-10 with the Euclidean loss (a) and with the cosine loss (b). Standard SimCLR uses the cosine loss. Colors as in Fig. 2. There was a similar difference in the $H$ space, but less pronounced.

initialized the new 2D output layer randomly, froze the entire network apart from this layer, and trained the last layer for 50 epochs (learning rate 0.12, no warm-up, no annealing). This ensured that the new output layer was reasonably aligned with the rest of the projection head. Afterwards, the entire model was unfrozen and trained for another 450 epochs (initial learning rate 0.12/1000, warm-up for 10 epochs, cosine annealing down to 0).

We experimented with two different options for 128D pre-training, either using standard SimCLR with the cosine loss, or using the Euclidean loss. We found that using Euclidean similarity for pretraining resulted in the final $t$-SimCNE loss 2.7 (Table 1, line 7), vs. 3.2 when using the cosine similarity (Table 1, line 6). Euclidean loss also resulted in visually more pleasing embedding with crisp and well-separated clusters (Fig. 2d vs. Fig. 2c). As the difference in the final loss was large, we believe that this training strategy is the optimal one (even though the final $k$NN accuracy was slightly lower compared to the cosine pretraining: 89% vs. 90%).

We confirmed this conclusion by applying $t$-SimCNE to CIFAR-100 (Fig. 3). We found that without any pretraining, $k$NN accuracy on the superclass level was 50% (Fig. 3a); with SimCLR pretraining using cosine loss, it was 65% (Fig. 3b); and with Euclidean pretraining, it was 68% (Fig. 3c). On the level of classes, the $k$NN accuracy was 33%, 47%, and 51%, respectively (Table A.1, lines 4–6).

Even though the cosine similarity led to a higher quality of the $H$ representation for both CIFAR-10 and CIFAR-100 (93% vs. 91% on CIFAR-10; Table 1, lines 1 vs. 3) compared to the Euclidean similarity; and even though the spherical constraint has theoretically appealing properties (Wang & Isola, 2020), the Euclidean similarity worked better for $t$-SimCNE pretraining. This may have to do with the norm distribution of the embedding vectors. Standard SimCLR with cosine loss has no discernible structure in the embedding norms (Fig. 4b), which is not surprising as the $\mathbf{z} \in \mathbb{R}^{128}$ vectors are normalized when computing the cosine similarity. But with Euclidean loss, the norms of $\mathbf{z}$ vectors strongly differ between classes (Fig. 4a). This leads to a reasonable 2D embedding even *before* the linear fine-tuning (Fig. A.4a and Fig. A.5a), and fine-tuning is able to work more effectively (Fig. A.4b,c and Fig. A.5b,c). Regarding dimensional collapse, it was less pronounced with Euclidean loss compared to the cosine loss (Fig. A.6).

## 3.5 ADDITIONAL EXPERIMENTS

We explored the effect of training budget on the resulting representations. The training strategy described above totaled 1500 epochs and took almost 20 hours on our hardware. Even when the training length was drastically reduced, pretrained models achieved a better performance than end-to-end training in 2D. With the total budget of 500 epochs ($400 + 25 + 75$), the final $k$NN accuracy was 86% (Table 1, line 9); with 1000 epochs in total ($775 + 25 + 200$), it was 89% (Table 1, line 8) — better than end-to-end training for 5000 epochs. Visually, all three budgets resulted in qualitatively similar embeddings (Fig. A.7).

To test the stability of $t$-SimCNE, we repeated the training procedure three times with different random seeds. We found that while the exact layout and cluster arrangement differed between runs, qualitatively the embeddings were very similar (Fig. A.8).

SimCLR itself could in principle be used for data visualization if the output dimensionality is set to 3, so that the embedding lies on the 2-sphere $S^2$. However, we found that this setup only resulted in $k$NN accuracy of 74% (Table 1, line 2) and the embedding itself looked poor (Fig. A.9).

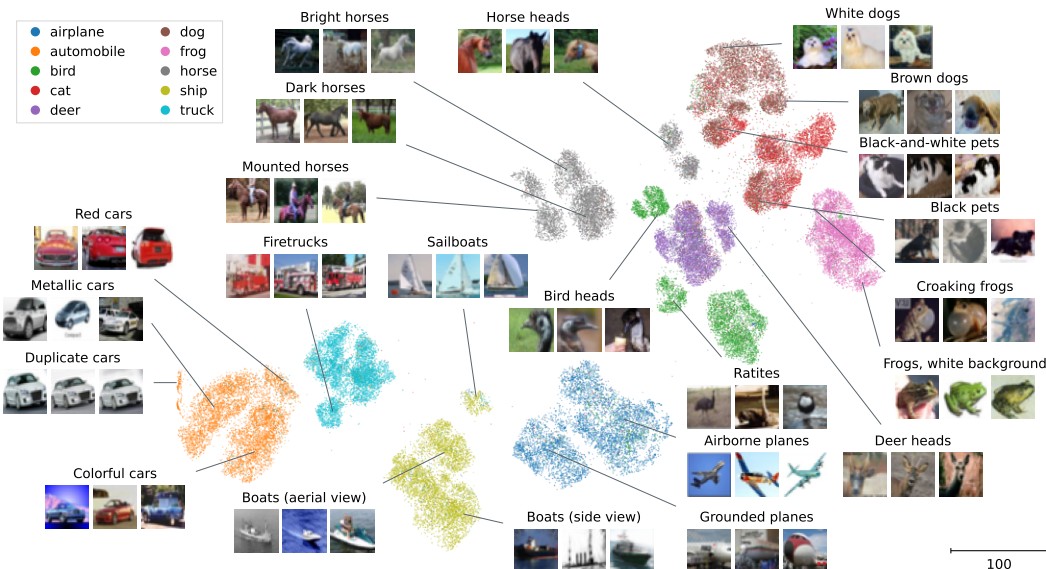

Figure 5: Annotated $t$-SimCNE embedding of the CIFAR-10 dataset. We manually annotated some of the prominent clusters by inspecting the images. Shown images are a random selection from the 15 nearest neighbors of the line tip.

# 4 Exploratory analysis shows the power of $t$-SimCNE

The ultimate goal of $t$-SimCNE is to be a tool for exploratory data analysis. In this section we demonstrate its power using the same CIFAR datasets as above.

## 4.1 $t$-SimCNE reveals subclass structure in CIFAR-10

We manually annotated the $t$-SimCNE embedding of CIFAR-10 for additional structure beyond the 10 originally defined classes (Fig. 5). We found that the embedding exhibited rich cluster structure, with many of the original classes (colors) splitting into several distinct clusters, or 'islands'.

We inspected the images within these clusters and found that they corresponded to meaningful semantic concepts. For example, within the 'ship' class, sailboats and other boat types differed clearly, such that sailboats formed an isolated cluster. Within the main 'ship' cluster, aerial view and sea level photographs varied systematically in their position in the 2D embedding. Similarly, the 'birds' class split into three well-separated islands: bird heads, ratites[1], and small birds. In the 'horse' class, horse heads and mounted horses formed separate clusters, and the remaining horse images were separated by color. Importantly, this within-class structure was not possible to infer from the original class labels alone, and we did not expect to find it when we started the project. Therefore, $t$-SimCNE helped us to discover additional latent structure in the dataset.

Although neighbor embedding methods are notoriously unreliable in preserving global structure of the data (Wattenberg et al., 2016), we found that $t$-SimCNE adequately represented some of the between-class structure. For example, the 'automobile' class was split into three main clusters: metallic cars, more colorful cars, and mostly bright red cars. The latter formed a separate cluster in the upper part of the orange island, next to one of the 'truck' clusters, consisting of bright red firetrucks.

One feature of the embedding that catches the eye, is a distinct, stripe-like orange cluster on the left (Fig. 5). Its oddly elongated and suspiciously dense shape suggests that it may be an artifact. Indeed, we found that this cluster consisted of near-duplicate pictures of the same three cars, such that three pictures appeared multiple times with only minor variations. A previous study focused specifically on near-duplicates in CIFAR datasets (Barz & Denzler, 2020) and identified 55 near-

---

[1]Large running birds such as ostriches and emus.

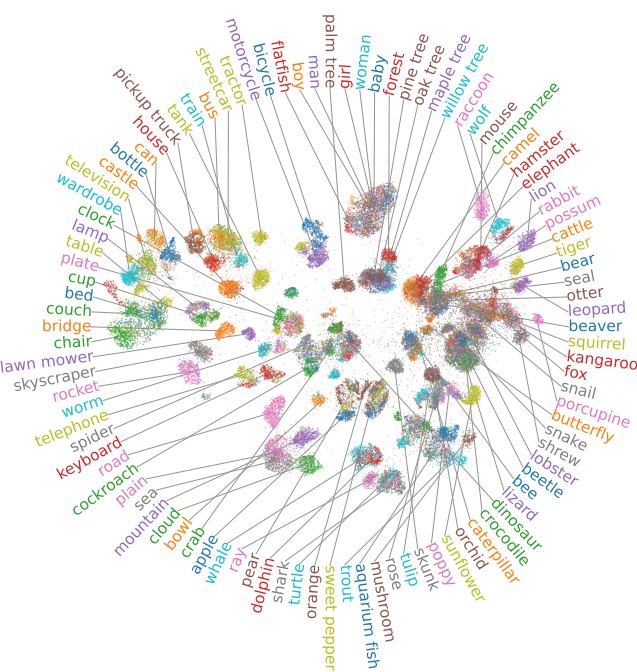

Figure 6: Annotated $t$-SimCNE embedding of the CIFAR-100 dataset. Class labels were positioned on the periphery in the order of $\text{atan2}(y, x)$ where $(x, y)$ is the mode of the kernel density estimate of embedding coordinates within each class.

duplicates in the 'automobile' class (Barz & Denzler, 2020, Table 1). With our exploratory analysis we found 163 almost identical images in the small orange island (Fig. A.10). Another study on labeling errors in standard computer vision datasets including CIFAR-10 missed the near-duplicates (Northcutt et al., 2021). This suggests that $t$-SimCNE is more intuitive and more sensitive than other, more conventional, ways to explore image datasets, and can be useful for quality control in actual practice.

The $t$-SimCNE visualization also highlighted what parts of the dataset SimCLR-like models struggle with. For example, the model seemed to be confused between dogs and cats as parts of these two classes were merged into one (upper-right corner: black-and-white pets, black pets). It is less noticeable, but within the cluster of airborne planes there were some images of birds. Those were pictures of birds flying in the sky, which exhibit a lot of visual similarity to airborne planes, hence the model's confusion.

While overall the embedding exhibited a pronounced cluster structure, there were a few outlier points located amidst the white space, which did not seem to belong to any one cluster. We investigated these images and found some obvious examples of outlier images. For example, one image, labeled as 'ship', depicts a person riding a jet ski (Fig. A.11c). Another image is labeled as 'truck', but we were unable to make out what was actually depicted (Fig. A.11f). These examples would be hard to find within a dataset of 60 000 images, but they clearly stick out in the 2D $t$-SimCNE visualization.

## 4.2 $t$-SIMCNE ELUCIDATES INTER-CLASS RELATIONS IN CIFAR-100

Similar observations apply to the $t$-SimCNE embedding of the CIFAR-100 dataset with its much more heterogeneous structure compared to CIFAR-10 (Fig. 6). In terms of the large-scale organisation, different animal species were placed mostly on the right side of the embedding, while man-made objects could be found on the left side. In terms of the fine-scale organization, the superclasses were separated very well in the embedding (Fig. 3c), as were many individual classes within a single superclass. For example, in the superclass 'flowers', the embedding clearly distinguished between orchids, sunflowers, roses, tulips, and poppy. Similarly, in the superclass 'large natural outdoor scenes', images of plain, sea, mountain, cloud, and forest were mostly non-overlapping.

On the other hand, in some other superclasses, individual classes were mixed together. For example, in the superclass 'people' (the topmost cluster), all five classes (boy, man, girl, woman, and baby) were intermingled, suggesting that the model struggled in this region of the feature space. Curiously, the same cluster also contained the 'flatfish' class. We found that the 'flatfish' images often show fishermen holding their catch, explaining their appearance in the 'people' island (Fig. A.12).

There were further examples of classes that $t$-SimCNE positioned next to a seemingly wrong superclass. The 'forest' class was separated from other members of the 'large natural outdoor scenes' superclass and placed next to the pine, oak, maple, and willow tree classes. This makes sense, as forest is close to trees both semantically and in terms of the image statistics.

These examples suggest that the class and the superclass definitions in the CIFAR-100 dataset do not always form a reliable ontology capturing all aspects of an image.

## 5 DISCUSSION

We developed a new self-supervised method, $t$-SimCNE, to visualize image datasets in 2D, and showed that it yields meaningful and interpretable visualizations of CIFAR-10 (Fig. 5) and CIFAR-100 (Fig. 6) datasets ($k$NN classification accuracy 89% and 51% respectively). We are not aware of any other unsupervised parametric methods that could yield comparably good 2D embeddings:

- $t$-SNE embeddings in pixel space are typically very poor as the Euclidean metric is not meaningful for image data (Fig. A.1). Any other neighbor embedding method, be it UMAP (McInnes et al., 2018), LargeVis (Tang et al., 2016), TriMap (Amid & Warmuth, 2019), etc., or any of their parametric versions (van der Maaten, 2009; Cho et al., 2018; Ding et al., 2018; Szubert et al., 2019; Kalantidis et al., 2022; Sainburg et al., 2021; Damrich et al., 2023), would have the same issue.

- $t$-SNE embeddings of the trained SimCLR representation are reasonably good (Fig. A.13), however (i) we found them qualitatively to be less interpretable than the $t$-SimCNE embeddings, as outliers, artifacts, and subclass structure were less noticeable; and (ii) they are not parametric, i.e. do not allow positioning out-of-sample images into an existing embedding, which is often relevant in applications.

- SimCLR with 3D output yields an embedding on 2-sphere $S^2$ but it was outperformed by $t$-SimCNE in terms of $k$NN accuracy and visual class separation, and also is cumbersome to visualize on a 2D plane as it requires a map projection (Fig. A.9).

- $t$-SNE embeddings of a trained ResNet-based classifier representation (Fig. A.14) are not unsupervised and not useful for data exploration, as they do not show much structure within each of the pre-defined classes.

- Finally, using a readily available off-the-shelf ResNet-based classifier pretrained on ImageNet and visualizing its representation with $t$-SNE (or UMAP, TriMap, etc.) is a very fast alternative approach to $t$-SimCNE. It is unsupervised in a sense that it does not use dataset labels (but is based on supervised ImageNet pretraining). While it is, again, not a parametric method, we found the resulting embeddings to have good $k$NN accuracy, especially for larger ResNets, such as ResNet-152 (Figs. A.15 and A.16). As it is much faster than $t$-SimCNE, this approach can make sense for initial exploration of the data, however we found that it is strongly outperformed by $t$-SimCNE in terms of visual class separation, as measured by the clustering-based adjusted Rand index (Fig. A.17).

Optimizing our architecture with 2D output was challenging and required a carefully tuned training setup. Our training scheme yielded good 2D embeddings, but led to the worse representation in the $H$ space compared to standard SimCLR. This suggests that it may be possible to further improve the training protocol and the fine-tuning schedule, which we leave for future work.

In the future it will be interesting to apply $t$-SimCNE to larger and more complex datasets, such as ImageNet (Russakovsky et al., 2015) or its subsets, e.g. Tiny ImageNet. It will also be interesting to apply it to real-life scientific data, e.g. in a biomedical domain. Our hope is that methods like $t$-SimCNE will be able to aid scientific exploration and discovery.

## ACKNOWLEDGMENTS

We thank Wieland Brendel and Evgenia Rusak for discussions on the topic.

This research was funded by the Deutsche Forschungsgemeinschaft (DFG, Germany's Research Foundation) via the Excellence Cluster 2064 "Machine Learning: New Perspectives for Science" (390727645), by the German Ministry of Education and Research (Tübingen AI Center, 01IS18039A), and by the Cyber Valley Research Fund (D.30.28739).

The authors thank the International Max Planck Research School for Intelligent Systems (IMPRS-IS) for supporting Jan Niklas Böhm.

## REPRODUCIBILITY

The code for generating our figures is available at `https://github.com/berenslab/t-simcne/tree/iclr2023`. Note that the runs on a GPU are not deterministic, due to nondeterminism introduced by optimized GPU kernels. Hence a perfect reproduction is impossible, but we confirmed that multiple runs yield very similar results. We always report the mean and standard deviation across several runs, and also show multiple visualizations in Fig. A.8.

We implemented a user-friendly class `TSimCNE` following the sklearn-style API; a minimal working example is shown in the Github repository. `TSimCNE.fit()` follows the setup described in this paper, save for one minor detail. We ran all experiments using dataset-level normalization of pixel values. After the experiments were completed, we realized that this normalization does not have any noticeable influence on the results, hence we omit it in the `TSimCNE` code. Future development of $t$-SimCNE will happen in the main branch of the Github repository: `https://github.com/berenslab/t-simcne`.

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

## APPENDIX A   SUPPLEMENTARY TABLES AND FIGURES

Table A.1: Model performance on the CIFAR-100 dataset. See Table 1 for description.

| # | Model | dim | Epochs | Linear in $H$ | $k$NN in $Z$ | Loss | Time (hr.) |
|---|-------|-----|--------|---------------|--------------|------|------------|
| 1 | Cosine (SimCLR) | 128 | 1000 | $66.7 \pm 0.5\%$ | $58.0 \pm 0.2\%$ | 5.8 | $12.0 \pm 0.1$ |
| 2 | Cosine | 3 | 1000 | $52.4 \pm 0.1\%$ | $15.7 \pm 0.2\%$ | 6.3 | $11.9 \pm 0.0$ |
| 3 | Euclidean | 128 | 1000 | $59.5 \pm 0.2\%$ | $54.5 \pm 0.1\%$ | 2.5 | $11.9 \pm 0.1$ |
| 4 | Euclidean | 2 | 1000 | $49.4 \pm 0.8\%$ | $33.2 \pm 0.2\%$ | 4.3 | $11.7 \pm 0.1$ |
| 5 | Cos.→Euclidean | 2 | 1500 | $65.1 \pm 0.1\%$ | $46.7 \pm 0.6\%$ | 3.6 | $17.5 \pm 0.2$ |
| 6 | **Eucl.→Euclidean** | **2** | **1500** | **$59.3 \pm 0.4\%$** | **$51.1 \pm 0.3\%$** | **2.9** | **$17.4 \pm 0.1$** |

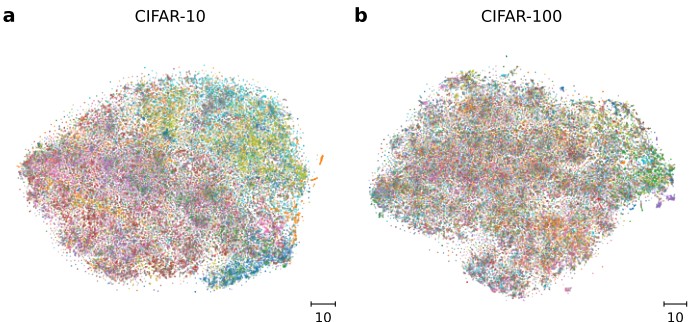

Figure A.1: $t$-SNE embedding of the CIFAR-10 (a) and CIFAR-100 (b) datasets in pixel space. Each image is $32 \times 32$ with three color channels, corresponding to vectors of dimensionality $32 \cdot 32 \cdot 3 = 3072$. They were embedded using openTSNE (Poličar et al., 2019) with default parameters. Colors correspond to classes as in Figs. 2 and 6. $k$NN accuracies were 33% and 13% (CIFAR-100 classes) in the embedding, and 33% and 15% when measured directly in the 3072-dimensional pixel space. For reference, our method $t$-SimCNE obtained 89% and 51% respectively.

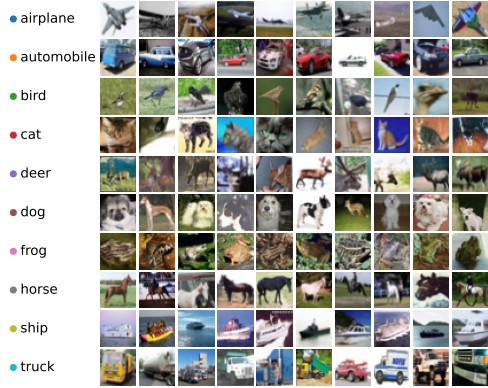

Figure A.2: Ten random images from each class of the CIFAR-10 dataset.

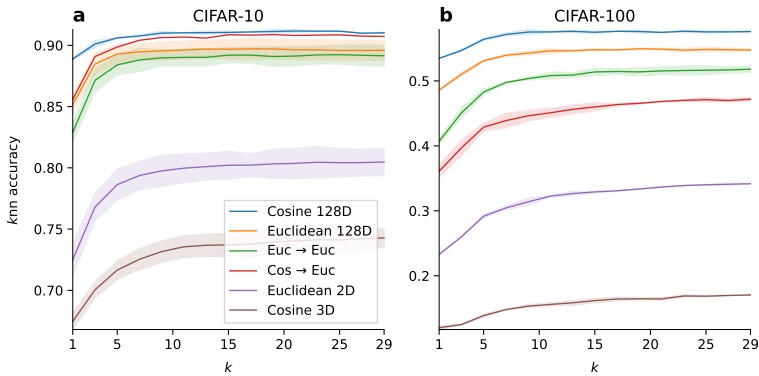

Figure A.3: $k$NN accuracy in the $Z$ space of various models listed in Tables 1 and A.1 as a function of $k$. Shading shows standard deviations across three runs. (a) CIFAR-10. (b) CIFAR-100.

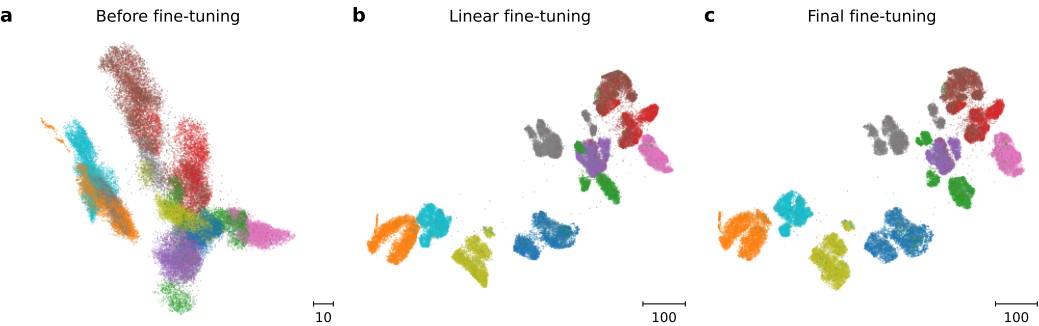

Figure A.4: Different stages of fine-tuning when training $t$-SimCNE on CIFAR-10. *(a)* The embedding after random initialization of the 2D output layer. *(b)* The embedding after training the output layer for 50 epochs, while the rest of the network stays frozen. *(c)* The final embedding after fine-tuning the entire network for 450 epochs. Colors as in Fig. 2.

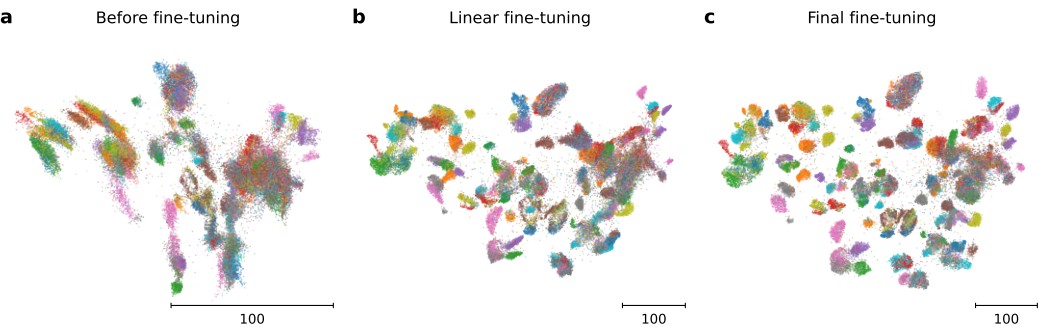

Figure A.5: Different stages of fine-tuning when training $t$-SimCNE on CIFAR-100. *(a)* The embedding after random initialization of the 2D output layer. *(b)* The embedding after training the output layer for 50 epochs, while the rest of the network stays frozen. *(c)* The final embedding after fine-tuning the entire network for 450 epochs. Colors as in Fig. 6.

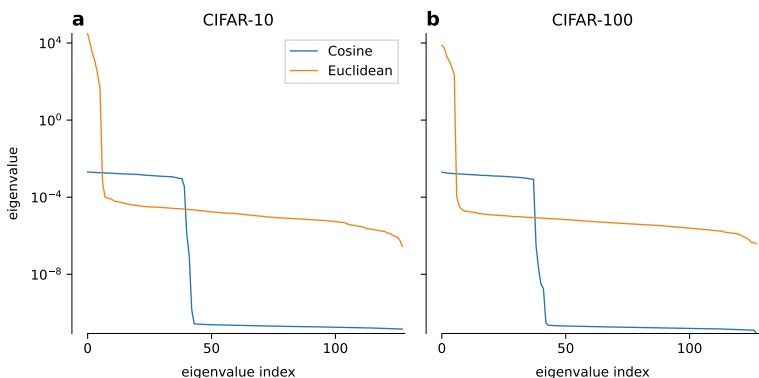

Figure A.6: Eigenvalue spectrum of the covariance matrix in the $Z$ space for 128-dimensional models trained with the cosine and with the Euclidean losses. (a) CIFAR-10. (b) CIFAR-100.

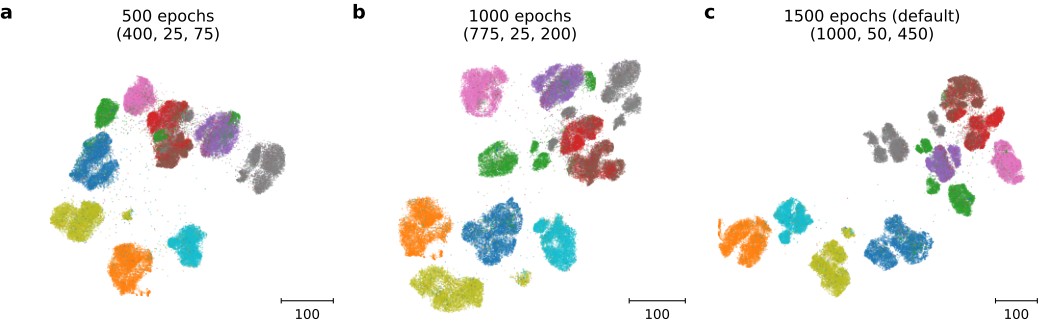

Figure A.7: $t$-SimCNE embeddings of CIFAR-10 dataset with different runtime budgets. *(a)* 500 epochs in total (400 for pretraining, 25 for the readout training, 75 for fine-tuning). *(b)* 1000 epochs in total (775 for pretraining, 25 for the readout training, 200 for fine-tuning). *(a)* 1500 epochs in total (1000 for pretraining, 50 for the readout training, 450 for fine-tuning). Embeddings were flipped along the $x$- and/or $y$-axis to maximize the alignment. Colors as in Fig. 2.

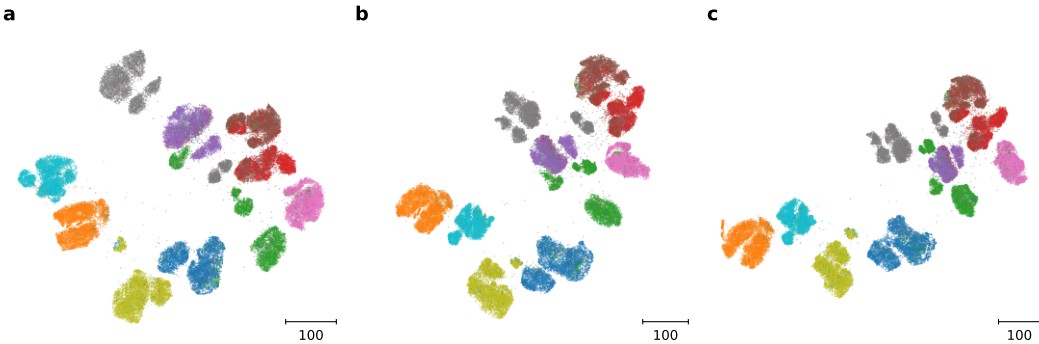

Figure A.8: $t$-SimCNE embeddings of CIFAR-10 dataset, trained from scratch with three different random seeds. Embeddings were flipped along the $x$- and/or $y$-axis to maximize the alignment. Colors as in Fig. 2.

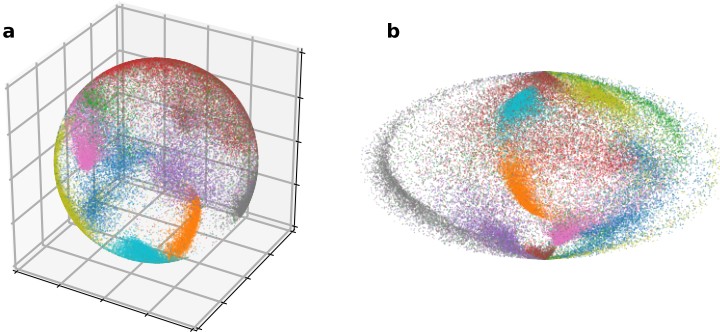

Figure A.9: 3D visualization of CIFAR-10 obtained with SimCLR with 3D output. *(a)* Embedding vectors on the unit sphere. *(b)* Hammer projection (equal-area) to two dimensions. Colors as in Fig. 2.

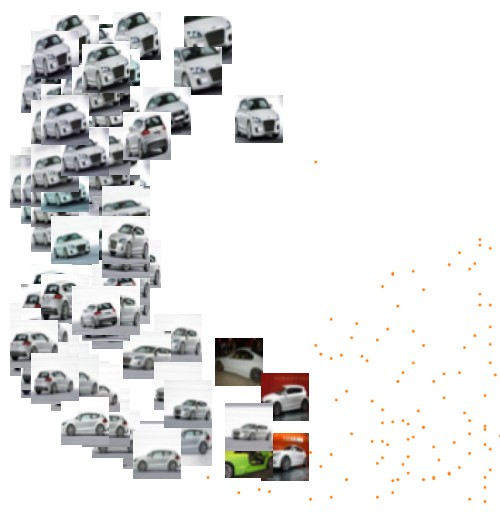

Figure A.10: Near-duplicates in CIFAR-10. A zoom-in into the $t$-SimCNE embedding. There are three distinct images, duplicated many times with small variations: (1) front view of a car, (2) rear view of a car, and (3) side view of a car. Colors as in Fig. 2. For this image the aspect ratio of the embedding has not been preserved.

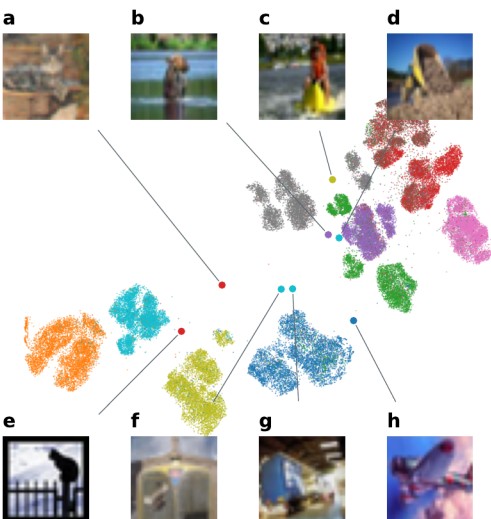

Figure A.11: Outliers in the $t$-SimCNE embedding of CIFAR-10. *(a)* A cat, almost blurred into the background. *(b)* An animal, labeled as 'deer', emerging from the water. *(c)* A person on a jet ski, labeled as 'ship'. *(d)* Mud in the foreground, and a truck unloading it in the background. *(e)* A cat silhouette. *(f)* Unknown, labeled as 'truck'. *(g)* Unknown, labeled as 'truck'. *(h)* Unknown, labeled as 'plane'. Colors as in Fig. 2.

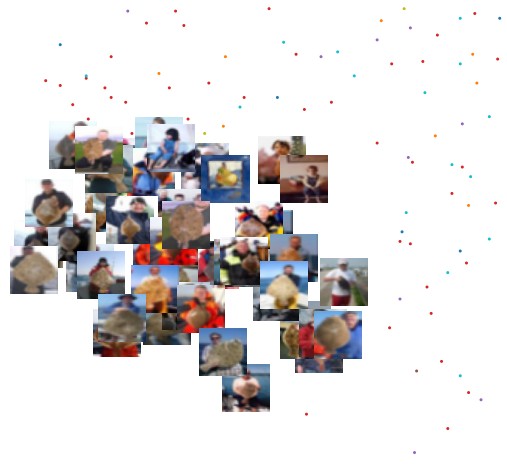

Figure A.12: Images of the class 'flatfish' in CIFAR-100 located next to the 'people' super-class in $t$-SimCNE embedding. Colors as in Fig. 6. For this image the aspect ratio of the embedding has not been preserved.

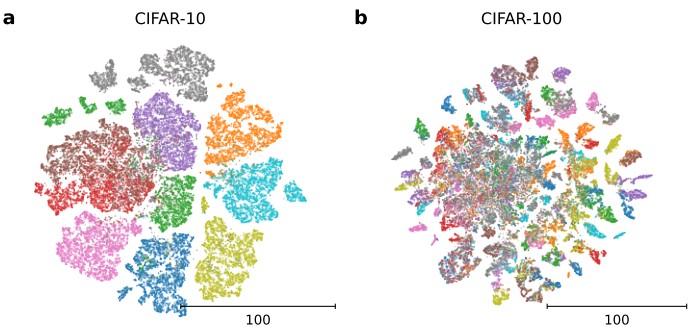

Figure A.13: $t$-SNE of the 512-dimensional standard (with cosine loss) SimCLR representation of CIFAR-10 (a) and CIFAR-100 (b). We used openTSNE (Poličar et al., 2019) with default parameters. Colors correspond to classes as in Figs. 2 and 6. The $k$NN accuracies were 91% for CIFAR-10 (a) and 57% for CIFAR-100 classes (b).

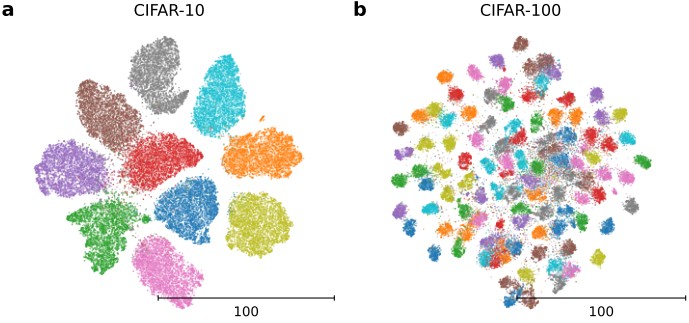

Figure A.14: $t$-SNE embedding of the CIFAR-10 (a) and CIFAR-100 (b) representations obtained with a ResNet-18 trained with the supervised cross-entropy loss on the same CIFAR data. The network architecture was the same as for the SimCLR experiments, but the projection head was mapping to ten (for CIFAR-10) or 100 (for CIFAR-100) softmax dimensions. We trained each network for 100 epochs, with initial learning rate 0.1 and linear annealing down to 0. We used the same set of data augmentations as we used for contrastive learning (Chen et al., 2020). Only training images were used for training; the test set accuracy after training was 92.8% for CIFAR-10 and 72.4% for CIFAR-100 classes. The ResNet output layer $H$ (see Fig. 1) was used as the input to $t$-SNE (both training and test images together). We used openTSNE (Poličar et al., 2019) with default parameters. Colors correspond to classes as in Figs. 2 and 6.

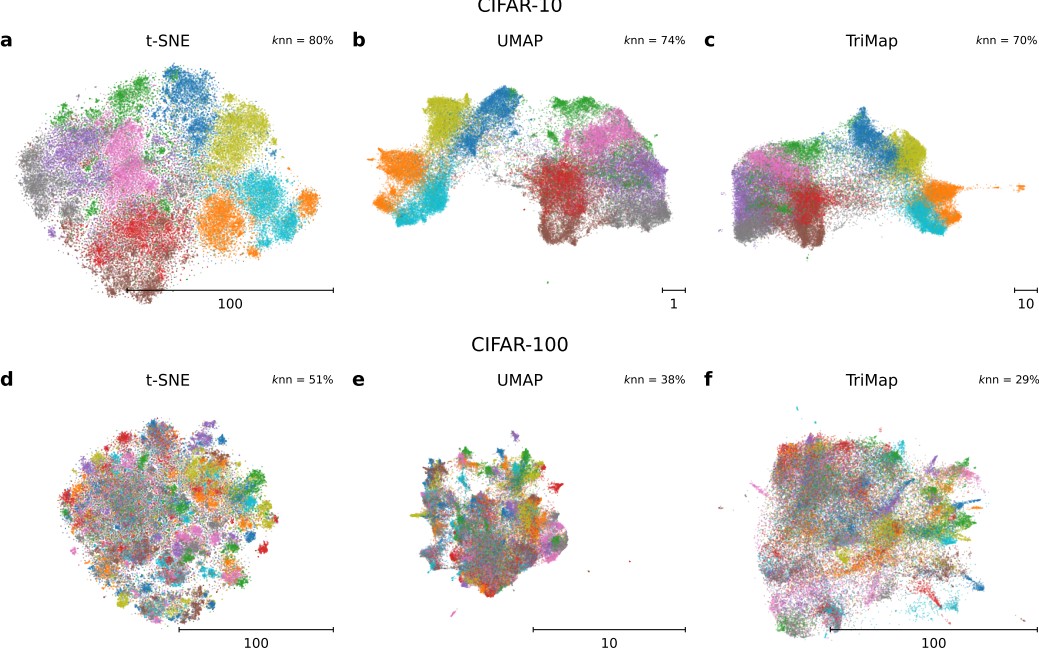

Figure A.15: Embeddings of the CIFAR-10 (a–c) and CIFAR-100 (d–f) representations obtained with a ResNet-18, pretrained on the ImageNet classification task. We used pretrained weights available in PyTorch (Paszke et al., 2019). The input images were first resized to $256 \times 256$ pixels and then center-cropped to $224 \times 224$ pixels, following He et al. (2016). The ResNet output layer $H$ (see Fig. 1) was used as the input to the visualization algorithms. We used openTSNE (Poličar et al., 2019), UMAP (McInnes et al., 2018), and TriMap (Amid & Warmuth, 2019) with default parameters. Colors correspond to classes as in Figs. 2 and 6. $k$NN classification accuracies are indicated in the corner of each panel (for CIFAR-100, it is the accuracy on the class level).

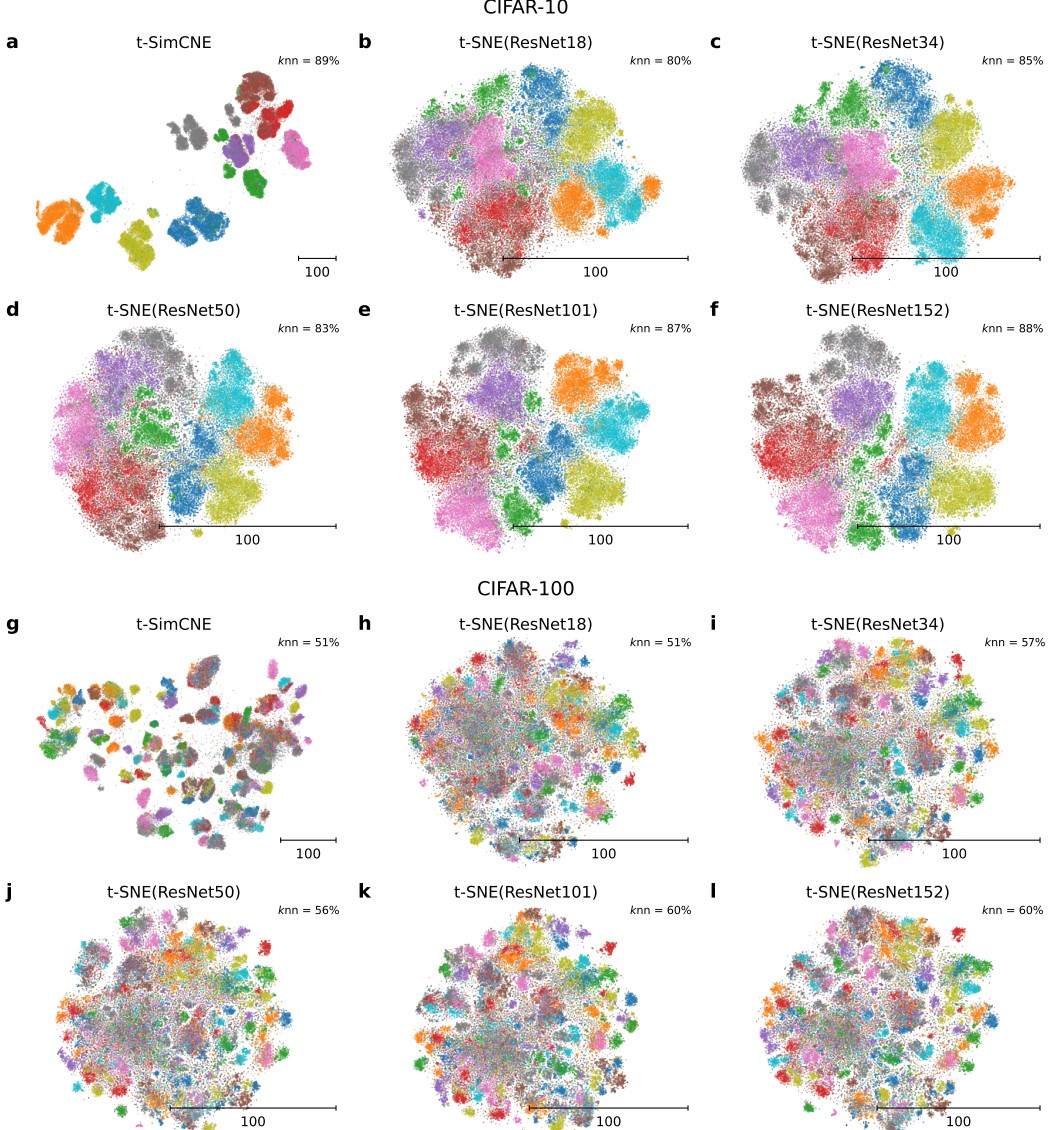

Figure A.16: Visualizations of $t$-SimCNE and of $t$-SNE applied to the representation obtained using ImageNet-pretrained ResNet networks of different size. *(a–f)* CIFAR-10. *(g–l)* CIFAR-100.

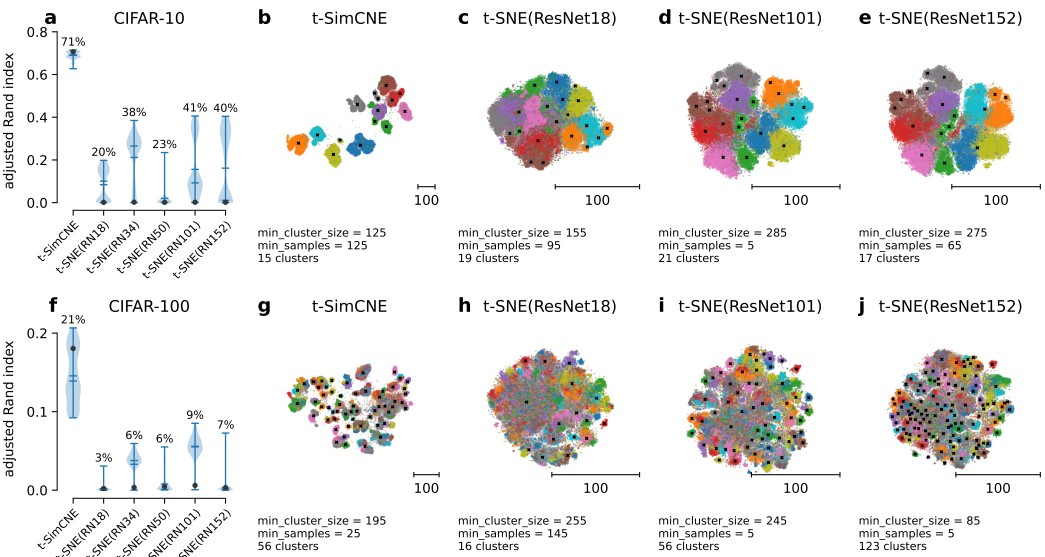

Figure A.17: Class separability comparison between $t$-SimCNE and $t$-SNE applied to the representation obtained using ImageNet-pretrained ResNet networks. We used clustering to assess how strongly classes were separated from each other in 2D. *(a)* The adjusted Rand index (ARI; Hubert & Arabie, 1985) between the clusters and the class labels on CIFAR-10. The clusters were obtained with HDBSCAN (McInnes et al., 2017) using the parameter grid `min_cluster_size` $\in \{5, 15, \ldots, 295\} \times$ `min_samples` $\in \{5, 15, \ldots, 145\}$ with the constraint `min_samples` $\leq$ `min_cluster_size`. The violin plots (Hintze & Nelson, 1998) show the distribution of the ARI values across different HDBSCAN parameter combinations, with the best value indicated above. The black dot denotes the ARI of the default HDBSCAN parameters (`min_cluster_size` = `min_samples` = 5). We used ResNet weights that are available in PyTorch. *(b–e)* The 2D visualizations of CIFAR-10 using $t$-SimCNE and $t$-SNE of pretrained ResNet representations. The cluster centroids corresponding to the best clustering (in terms of ARI) are shown as black cross marks, and the corresponding HDBSCAN parameters are given below. *(f–j)* The same for CIFAR-100.

