# OpenReview forum: "Unsupervised visualization of image datasets using contrastive learning"
_ICLR.cc/2023/Conference — ICLR 2023 poster_

### Official Review · Reviewer_LCjY · 2022-10-21

**Confidence:** 3
**Correctness:** 3
**Technical Novelty And Significance:** 3
**Empirical Novelty And Significance:** 3
**Recommendation:** 6

**Clarity, Quality, Novelty And Reproducibility:**

It is clear, the research is of fine quality and, as previously mentioned, technical novelty, even if somehow marginal, is still of interest to the data analyst with a taste for exploratory visualization. The availability of the code and data makes the results reproducible.

**Strength And Weaknesses:**

Strengths:
Quite well written.
Good discursive didactic style.
Marginal but useful technical novelty.
Excellent review of existing literature.
Code available.

Weaknesses:
Marginal/incremental novelty
A bit too much setting-cooking, which arises some doubts as to the generalizability of the proposal.

**Summary Of The Paper:**

Interesting paper concerning unsupervised visualization of image data using neighbour embeddings that borrows ideas of ideas from contrastive learning.
It is motivated by the need of mapping from the high-dimensional pixel image  space into lower dimensions (two, in this study), where, as the authors pose it, "distances in pixel space are often not capturing our sense of similarity", so that "neighbors are not semantically close".

**Summary Of The Review:**

Interesting paper concerning unsupervised visualization of image data using neighbour embeddings that borrows ideas of ideas from contrastive learning.
It is motivated by the need of mapping from the high-dimensional pixel image  space into lower dimensions (two, in this study), where, as the authors pose it, "distances in pixel space are often not capturing our sense of similarity", so that "neighbors are not semantically close".
The paper is quite well written and its structure is quite fresh and didactic.
It uses contrastive learning SimCLR method that generates high-dim image representations and it is used to directly optimize a 2D embedding.
The main novel contribution is the pretraining strategy required for this proposed model to work, specially given that two other recent papers seem to be chasing the same idea (Zang et al., Hu et al., both from this year) Such pretraining strategy is a bit too trial and error for my taste (in the sense that it makes you wonder whether it should be tailored to the analyzed dataset at hand), althoug, arguably, understandably so.
The literature background section, by the way, is excellent.
In section 3.3, for embedding quality assessment, authors use kNN and indicate that "For kNN accuracy, we used the scikit-learn [...] implementation with k = 15". Why k=15? What's the rationale for that choice? Did authors investigate other alternative values of k?
The experimental body, even if not being fully conclusive, provides very good evidence of the data exploratory potential of the model.

Other:
It would be useful to clarify in the introduction that Figures referenced as A.x refer to an appendix.
Figure 1 should be better located at the end of the introduction.
p.2, section 2: "In this work, we are interested in an unsupervised but parametric mapping that ALLOWS embedding ..."

---

> ### Author Response · Authors · 2022-11-14
> **Response to Reviewer LCjY**
>
> We thank the Reviewer for the positive comments!  We have fixed the typo in the manuscript and would like to respond to three further points.
>
> > Such pretraining strategy is a bit too trial and error for my taste (in the sense that it makes you wonder whether it should be tailored to the analyzed dataset at hand), although, arguably, understandably so.
>
> This is a fair point. As we showed, the naive approach of directly optimizing the 2D embedding yields only a poor result -- and indeed we have spent considerable effort coming up with a more viable training strategy. However, the strategy that we describe has only been tweaked using CIFAR-10, and we were happy to see that it worked directly for CIFAR-100 without further modifications.
>
> The 'trial and error' feel may to some extent be caused by our description which is largely chronological. We could have presented the best approach first and describe the rest as 'ablation experiments', but opted for a more honest chronological exposition.
>
> > In section 3.3, for embedding quality assessment, authors use kNN and indicate that "For kNN accuracy, we used the scikit-learn [...] implementation with k = 15". Why k=15? What's the rationale for that choice? Did authors investigate other alternative values of k?
>
> We used $k=15$ throughout all experiments because it typically works well in the neighbour embedding domain (e.g. for t-SNE or UMAP). In response to this comment, we computed kNN accuracy for values of k between 1 and 30, and found that the results stayed qualitatively the same for any value of $k$. This is now shown in the supplementary Figure A.3.
>
> > It would be useful to clarify in the introduction that Figures referenced as A.x refer to an appendix.
>
> We have made this more explicit by referring to Figure A.x as Supplementary Figure A.x when we first reference an appendix figure.

---

### Official Review · Reviewer_nGad · 2022-10-24

**Confidence:** 4
**Correctness:** 4
**Technical Novelty And Significance:** 3
**Empirical Novelty And Significance:** 4
**Recommendation:** 10

**Clarity, Quality, Novelty And Reproducibility:**

This paper is very clearly written. I find the reasoning and plots convincing. The method is novel as far as I know. I did not check, but it seems that the results are reproducible.

**Strength And Weaknesses:**

For unsupervised visualization of image data, one challenge is how to provide as much information as possible in a 2D space. By utilizing the t-SNE loss function in the neighbor embedding literature, the authors propose a natural way to embed and visualize images, which they call t-SimCNE.

**Strengths:** It is clearly shown that it is nontrivial to train the model with a loss similar to t-SNE. Moreover, the authors provide several pre-training strategies and demonstrate that their pretraining/fine-tuning with Euclidean loss strategy works the best. See Figure 2 and 3.

Furthermore, it is clearly shown that data visualization t-SimCNE displays fine-grained structures that is very useful for data diagnostics and interpretation.

**Weaknesses:** I do not find major weaknesses. Perhaps the authors make comments on (1) if there is dimensional collapse issue during training, especially if we do not follow the proposed training strategies, and (2) what information is lost when embedded data is restricted to 2D instead of the initial 128D.

**Summary Of The Paper:**

In recent years, neighbor embedding methods, such as t-SNE and UMAP, are popular for visualizing high-dimensional data. Motivated by the success of contrastive learning, the authors propose a new method t-SimCNE for unsupervised visualization of image data. The proposed method builds on the popular SimCLR, but replaces the loss with a heavier-tail loss as is used in t-SNE, because it is better for visualization 2D representation.

The authors detailed training strategies for training/pre-training the new model, showing that the proposed strategies are crucial for resolving nontrivial implementation issues. The authors also present thorough experimental results, showing that t-SimCNE is good at showing revealing fine-grained representation information and subclass structure.

**Summary Of The Review:**

The authors provide a very useful method for visualizing image data by combining SimCLR and t-SNE in a nontrivial way (implementation-wise). I think that this paper has potential impact, and would highly recommend acceptance of this paper.

---

> ### Author Response · Authors · 2022-11-14
> **Response to Reviewer nGad**
>
> We thank the Reviewer for the very positive review and are happy about the encouraging comments! We would like to respond to two points.
>
> > (1) if there is dimensional collapse issue during training, especially if we do not follow the proposed training strategies
>
> For the models with 2D output there is no dimensional collapse, as can be seen directly in the visualizations (dimensional collapse in this case would mean that the embedding is constrained to 1 dimension).
>
> However, in response to this comment we looked at the eigenspectra of 128D embeddings trained with the cosine and the Euclidean losses. We found that the cosine loss leads to a pronounced collapse, with eigenvalues sharply dropping to 10^-9 after around 40. With Euclidean loss, there was a sharp drop to 10^-4 after only 6 eigenvalues but the remaining eigenvalues were decaying very slowly and did not show further collapse. This is now shown in  supplementary figure A.7. We believe this difference requires further investigation (as does the issue of dimensional collapse in general).
>
> > (2) what information is lost when embedded data is restricted to 2D instead of the initial 128D.
>
> This is an interesting question and we have been wondering about it as well. kNN accuracy drop from 128D to 2D provides a numerical assessment of the information loss. For CIFAR-10, the drop was only 2% (see Table 1). We believe this is negligible, and means that for CIFAR-10 there was almost no loss of information. For CIFAR-100, the drop was 6% (from 57% to 51%, see newly added supplementary Table A.1). This is a more considerable drop, and we are currently thinking about useful ways of studying it in more detail (e.g. which entries of the confusion matrix decreased particularly strongly).

---

### Official Review · Reviewer_25XC · 2022-10-24

**Confidence:** 4
**Correctness:** 4
**Technical Novelty And Significance:** 2
**Empirical Novelty And Significance:** 2
**Recommendation:** 5

**Clarity, Quality, Novelty And Reproducibility:**

It is unclear to me what do $i$, $j$ (i.e., the similar pair) correspond to in Eq (4)? TriMap uses the k-NN of each point $i$ to draw $j$ (the similar point), and point $k$ (dissimilar point) is drawn randomly from the remaining points that are farther away than $j$. In the proposed method, do you create new augmentations for the 2D embedding? I think this needs to be clarified further.

**Strength And Weaknesses:**

I'm surprised that the authors have missed an important contrastive-loss-based dimensionality reduction method, called TriMap [1], that relies on a very similar principle for generating 2D embeddings. The proposed method can be seen as a variant of TriMap where the input representation is generated by a ResNet.

[1] Ehsan Amid and Manfred K. Warmuth. "TriMap: Large-scale dimensionality reduction using triplets." arXiv preprint arXiv:1910.00204 (2019).



**Summary Of The Paper:**

The paper proposes a method for unsupervised visualization of image datasets. The proposed method relies on first training a NN using contrastive learning and using its representation to minimize another contrastive loss to project the embedding to 2D. The results are validated on standard image datasets.

**Summary Of The Review:**

The proposed approach has a significant amount of overlap with TriMap, a contrastive-loss-based dimensionality reduction method. The authors need to clarify how their method differs from TriMap, what advantages (if any) it provides, and perform a comparison.


UPDATE: I'm increasing my score to acknowledge the authors' efforts to include additional results. However, I still believe the contributions of the paper are very limited. Please see the response to the authors' feedback.

---

> ### Author Response · Authors · 2022-11-11
> **Response to Reviewer 25XC**
>
> We thank the Reviewer for their time, but there seems to be some confusion here. Our proposed method, t-SimCNE, has very little similarity to TriMap.
>
> > I'm surprised that the authors have missed an important contrastive-loss-based dimensionality reduction method, called TriMap [1], that relies on a very similar principle for generating 2D embeddings.
>
> The reviewer is right that TriMap uses a contrastive-based triplet loss which bears similarity to the InfoNCE loss (employed by us). But TriMap relies on the standard kNN graph, whereas we follow the ‘contrastive learning’ approach that relies on random image augmentations to generate pairs of ‘neighbours’. This makes the two methods entirely different!
>
> > The proposed method can be seen as a variant of TriMap where the input representation is generated by a ResNet.
>
> This claim by the reviewer is incorrect. Our method does not rely on a pre-trained network, it trains ResNet representation and the 2D mapping from scratch. Whereas running TriMap on top of a pretrained ResNet would necessarily rely on a fixed pretrained representation.
>
> That said, we now performed this experiment and applied TriMap (together with UMAP and t-SNE) to the CIFAR-10 and CIFAR-100 representations obtained with a pretrained ResNet18, available off-the-shelf in Pytorch library. The results are shown in Figure A13 in the updated paper. All of the resulting embeddings are much poorer than our t-SimCNE embeddings, and in particular TriMap yields kNN accuracy 70% on CIFAR-10 and 29% for CIFAR-100 (vs. 89% and 51% with our method).
>
> We now refer to TriMap and to this new figure in the updated Discussion.
>
> > It is unclear to me what do i,j, (i.e., the similar pair) correspond to in Eq (4)? [...] In the proposed method, do you create new augmentations for the 2D embedding?
>
> The notation was introduced half a page before Eq (4) where we write: “Here, indices $i$ and $j$ correspond to two data augmentations of the same original image”.

---

> > ### Comment · Reviewer_25XC · 2022-11-11
> > **Thank you for your response**
> >
> > Thank you for your timely response and clarification. Also, thank you for including the updated results.
> >
> > I agree that the results of t-SimCNE using joint training are much better than applying UMAP and TriMap on the pre-trained representations. However, I believe you should position yourself better in terms of the contributions of the work:
> > 1) Contrastive pre-training of neural networks using data augmentations isn't new (Chen et al. 2020).
> > 2) Using a network for improving the representation of DR methods has a long history, starting with Hinton and Salakhutdinov (2006). Many follow-up papers combine a deep network with a loss function for 2D embedding on top for DR.
> > 3) Contrastive learning for DR (although on the k-NN graph), i.e., TriMap, is also not new. In fact, there are methods like ivis (Szubert et al. 2019, https://github.com/beringresearch/ivis) or TLDR (Kalantidis et al., 2021) that combine a deep network feature extractor with a contrastive loss for DR.
> >
> > So in summary: the main novelties of the paper are switching from a deep autoencoder to a ResNet model and using data augmentations for contrastive learning. I believe the authors should review the previous work better and clarify what their approach offers. I'm increasing my score to acknowledge the authors' efforts.
> >
> >
> > * Geoffrey E. Hinton, and Ruslan R. Salakhutdinov. "Reducing the dimensionality of data with neural networks." Science 313, no. 5786 (2006): 504-507.
> > * Benjamin Szubert, Jennifer E. Cole, Claudia Monaco, and Ignat Drozdov. "Structure-preserving visualisation of high dimensional single-cell datasets." Scientific reports 9, no. 1 (2019): 1-10.
> > * Yannis Kalantidis, Carlos Lassance, Jon Almazan, and Diane Larlus. "Tldr: Twin learning for dimensionality reduction." arXiv preprint arXiv:2110.09455 (2021).

---

> > > ### Author Response · Authors · 2022-11-14
> > > **Literature**
> > >
> > > Thank you for acknowledging our efforts and for the pointers to the literature! Of course we are not claiming that our work has no precedents. We do, however, believe that there is no other method that allows to train a mapping from pixel space to 2D that would achieve comparable classification accuracy (on CIFAR-10 and 100) to our method.
> > >
> > > Regarding the papers that the you mentioned:
> > >
> > > * Chen et al. 2020 is currently cited in our paper eight times; it is difficult to make it more prominent.
> > >
> > > * Szubert et al. 2019 and Kalantidis et al., 2021 -- thank you for these references. These are parametric neighbour embedding methods, i.e. they train a neural network to preserve kNN graph. Methods like parametric UMAP (Sainburg et al. 2022) or parametric t-SNE (van der Maaten 2009; Damrich et al. 2022; also net-SNE of Cho et al. 2018 and scvis of Ding et al. 2018) also fall into this category. All of these methods are based on the kNN graph and hence are unsuitable for visualizing image datasets like CIFAR. We have now inserted these citations and some comments into the Discussion.
> > >
> > > * Hinton & Salakhutdinov 2006 is a seminal paper that suggested to use autoencoders with 2D bottleneck for visualization. Note that they used a non-convolutional architecture and did not employ data augmentations while training, so their method is not tailored to image data and would not be able to produce satisfactory embeddings of CIFAR-10/100 datasets. An autoencoder approach could in principle work for 2D visualizations of image data, however we are not aware of any more recent research in this direction (using modern deep learning architectures, data augmentations, etc.). If the Reviewer is aware of autoencoder methods with available public code tailored for 2D visualization of image data, we would be happy to try them on CIFAR-10/100 and benchmark against our t-SimCNE.

---

### Official Review · Reviewer_TNpm · 2022-10-25

**Confidence:** 4
**Correctness:** 3
**Technical Novelty And Significance:** 3
**Empirical Novelty And Significance:** 4
**Recommendation:** 6

**Clarity, Quality, Novelty And Reproducibility:**

The paper is well-written, and all hyper-parameters to reproduce the method seem to be included. While there is some previous work that discussed the relationship between SimCLR and tSNE, no derivations or empirical results have been provided to date (so the contribution of the paper is novel). While there are some key questions in the weakness section that need to be clarified, overall, the paper would be a useful contribution.


**Strength And Weaknesses:**

Strengths
- A novel application of self-supervised learning, dimensionality reduction, is discussed and explored.
- Paper is clearly written
- A mathematical relationship between the SNE and simCLR loss is derived
- Ablation study of the different training strategies (number of training epochs, type of loss, type of training/fine-tuning) elucidates when does the proposed method works and does not work
- Experiments are conducted on two standard public image benchmarks (CIFAR-10 and CIFAR-100 datasets)
- Results are compelling and convincing

Weaknesses
- There is no computational analysis provided. tSNE is known as a relatively lightweight technique that can be used to quickly browse a large dataset. It is not clear whether the proposed method, t-SimCNE, can be run in a similar amount of time and amount of computational resources (batch size of 1024 is large and requires high-memory GPU). A discussion of when the proposed method would be more suitable, in comparison to tSNE, is missing.
- The key weakness of tSNE pointed out in the intro is that it does not work well on large datasets (reader is pointed to figure A.1). However, in Figure A1, it is not clear what the problem is? Perhaps a zoom in to the graph to illustrate the mix of the labels would help
- In Equation (3), a relationship between the stochastic neighbor embedding (SNE) and the SimCLR loss is derived. What is the relationship between SNE and t-SNE?
- What happens if the pre-training on SimCLR is done on a different dataset (e.g., ImageNet), as a strategy to save training time?


**Summary Of The Paper:**

The paper reasons about the relationship between SimCLR, a recent self-supervised learning method, and tSNE, a popular dimensionality technique for data visualization. At the intersection of these two methods, the paper proposed a new method, based on self-supervised learning, for high-dimensional data visualization (to be used instead of tSNE). The new method, known as t-SimCNE, removes the 1-norm constraint on the standard SNE loss, replacing cosine with Euclidean loss, and trains the SimCLR network in two stages to achieve better performance. Compelling results with respect to tSNE are demonstrated in the CIFAR-10 and CIFAR-100 datasets, both qualitative and accuracy.


**Summary Of The Review:**

The paper offers a novel and thorough empirical and mathematical analysis of tSNE and SimCLR and derives a learning-based dimensionality reduction method.

EDIT: After reading the response and comments by the other reviewers, I believe that the contribution is marginal. The paper would benefit from a more thorough discussion and comparisons to prior work, as well as ablation studies with different backbone settings. I am also concerned about the large runtime of the method, which might prevent it from being used more widely in practice.

---

> ### Author Response · Authors · 2022-11-12
> **Response to Reviewer TNpm**
>
> We thank the Reviewer for the clear summary and positive review of our work. To address the concerns, we added numerical evaluation of the pixel-space representation and extended the discussion.
>
> > There is no computational analysis provided. tSNE is known as a relatively lightweight technique that can be used to quickly browse a large dataset. It is not clear whether the proposed method, t-SimCNE, can be run in a similar amount of time and amount of computational resources (batch size of 1024 is large and requires high-memory GPU). A discussion of when the proposed method would be more suitable, in comparison to tSNE, is missing.
>
> We thank the reviewer for these remarks and point him/her to Table 1, where we reported the runtimes of our methods already in the original paper, which might have been overlooked by the reviewer. On our hardware (which was a standard GeForece RTX 2080 Ti), one run of t-SimCNE on CIFAR-10 took around 19 hours. We showed that reducing the overall number of epochs from 1500 to 500 reduces the time to 6 hours, while still producing a sensible visualization with high kNN accuracy (Figure A5).
>
> Of course the reviewer is right, and even 6 hours is much more than t-SNE, which takes ~1 minute. However, t-SNE in pixel space produces a meaningless embedding (see below). A more sensible approach is to apply t-SNE on the representation obtained with a pre-trained ResNet, as we do in Figure A13. This is not much slower than t-SNE, and can make sense for a quick initial exploration of the data. Still, as we show in Figure A13, t-SimCNE embeddings are much more interpretable and insightful. We have now added some of these considerations to the Discussion.
>
> > The key weakness of tSNE pointed out in the intro is that it does not work well on large datasets (reader is pointed to figure A.1). However, in Figure A1, it is not clear what the problem is? Perhaps a zoom in to the graph to illustrate the mix of the labels would help.
>
> Thank you for the suggestion. To more clearly illustrate the problem with tSNE in the pixel space, We now computed the kNN accuracy in the high-dimensional pixel space, which is 33% for CIFAR-10 and 15% for CIFAR-100. This shows that the pixel-space neighbors are not semantically close. For the 2D layout shown in Figure A1, the kNN accuracy only slightly dropped to 33% and 13% for the two datasets. We have now added this information to the caption of Figure A1. For comparison, kNN accuracies are 89% and 51% for our method, underscoring its superiority.
>
> Note that it is not the size of the dataset that is a problem here (t-SNE can work well on much larger datasets), but the fact that pixel-space neighbors are not semantically close.
>
> > In Equation (3), a relationship between the stochastic neighbor embedding (SNE) and the SimCLR loss is derived. What is the relationship between SNE and t-SNE?
>
> The only difference between SNE and t-SNE is that SNE uses $\exp(-d^2)$ similarity between embedding vectors while t-SNE uses $1/(1+d^2)$ similarity; here $d$ refers to the Euclidean distance in the embedding. In the manuscript, this is explained a few paragraphs after Equation (3), in Section 3.2.
>
> > What happens if the pre-training on SimCLR is done on a different dataset (e.g., ImageNet), as a strategy to save training time?
>
> This is an excellent question. We have not experimented with this because we did not want to rely on any existing  representations and aimed to train everything from scratch. In practice, it is likely that using publicly available  pretrained weights could speed up the convergence and the fine-tuning. We could even use ResNet weights from supervised pretraining on ImageNet, as available within the Pytorch library, and fine-tune later. While we will certainly experiment with this approach in the future, there wasn't sufficient time to do this during the revision period.

---

> > ### Comment · Reviewer_TNpm · 2022-11-15
> > **thank you**
> >
> > Thank you for your responses. I have concerns regarding the relationship of the proposed work highlighted by other reviewers and the ability of the method to be used in practice, due to large runtime (see comment in review), therefore, I will keep my rating.

---

### Decision · Program_Chairs · 2023-01-20

**Decision:**

Accept: poster

**Justification For Why Not Higher Score:**

The authors claim that their contribution/aim is developing a high-dimensional data visualization method, which makes the rationality of the proposed method becomes questionable. As the reviewers pointed out, the complexity of the proposed t-SimCNE is much higher than the classic t-SNE method, which requires time-consuming two-step (pretraining and fine-tuning) strategies on GPUs. Additionally, most existing visualization methods are transductive, while the proposed method is inductive. Whether the proposed method is robust to new samples should be discussed.

Furthermore, I checked another related submission suggested by the authors (https://openreview.net/forum?id=XFSCKELP3bp). As the authors mentioned, that work explores the relationship between t-SNE and SimCLR. In my opinion, however, that work is more solid --- besides providing analysis about SNE and contrastive learning, that submission mainly considers improving contrastive learning via SNE. Additionally, it does not focus on the 2D visualization task proposed by this submission, so the concerns I mentioned above are not important for that work.

**Justification For Why Not Lower Score:**

All the reviewers agree that visualizing image data with the help of contrastive learning-assisted t-SNE is interesting, and one reviewer even gave 10 points to this submission.

**Metareview: Summary, Strengths And Weaknesses:**

The authors propose an interesting idea combining contrastive learning with t-SNE. Experimental results show that the proposed method successfully explores the clustering structure of high-dimensional data, demonstrating the proposed method's usefulness to some degree.

Strengths:
(1) The idea and the corresponding derivation are clear and easy to follow.
(2) Analytic experiments verify the performance of the proposed method to some degree.

Weaknesses:
(1) As a visualization method, the efficiency of the proposed method is not convincing, which may limit its application in practice.

**Note From Pc:**

if the above contains the word "oral" or "spotlight" please see: "oral" presentation means -> notable-top-5% and "spotlight" means -> notable-top-25%. As stated in our emails, we are disassociating presentation type from AC recommendations

**Summary Of Ac-Reviewer Meeting:**

After the rebuttal phase, the scores of the paper are 6, 6, 10, 5. All the reviewers agree that this is an interesting and important analytic work, which may be helpful for visualizing image datasets with better clustering structures. However, as a visualization method, the efficiency is questionable, as some reviewers mentioned.

I think a virtual meeting is necessary. My time zone is UTC+8.